# Human Psychophysiological Activity Estimation Based on Smartphone Camera and Wearable Electronics

**Alexey Kashevnik** [1],*, **Mikhail Kruglov** [2], **Igor Lashkov** [1], **Nikolay Teslya** [1], **Polina Mikhailova** [2], **Evgeny Ripachev** [2], **Vladislav Malutin** [2], **Nikita Saveliev** [2] and **Igor Ryabchikov** [2]

[1]  Computer Aided Integrated Systems Laboratory, SPIIRAS, St. Petersburg 199178, Russia; igla@iias.spb.su (I.L.); teslya@iias.spb.su (N.T.)
[2]  Information Technology and Programming Faculty, ITMO University, St. Petersburg 197101, Russia; krumih@mail.ru (M.K.); p.mikhaylova@icloud.com (P.M.); e.ripachev@gmail.com (E.R.); vladmalsims@gmail.com (V.M.); nikitasavelev11@gmail.com (N.S.); i.a.ryabchikov@gmail.com (I.R.)
*  Correspondence: alexey.kashevnik@iias.spb.su; Tel.: +7-812-328-8071

**Abstract:** This paper presents a study related to human psychophysiological activity estimation based on a smartphone camera and sensors. In recent years, awareness of the human body, as well as human mental states, has become more and more popular. Yoga and meditation practices have moved from the east to Europe, the USA, Russia, and other countries, and there are a lot of people who are interested in them. However, recently, people have tried the practice but would prefer an objective assessment. We propose to apply the modern methods of computer vision, pattern recognition, competence management, and dynamic motivation to estimate the quality of the meditation process and provide the users with objective information about their practice. We propose an approach that covers the possibility of recognizing pictures of humans from a smartphone and utilizes wearable electronics to measure the user's heart rate and motions. We propose a model that allows building meditation estimation scores based on these parameters. Moreover, we propose a meditation expert network through which users can find the coach that is most appropriate for him/her. Finally, we propose the dynamic motivation model, which encourages people to perform the practice every day.

**Keywords:** meditation estimation; neural networks; human behavior patterns

## 1. Introduction

The growing tension and stress of daily life is becoming a problem nowadays. This leads to the fact that people worldwide are paying more and more attention to their physical activities. At a scientific conferences on psychology, the problem of stress and depression was discussed with scientists, highlighting that stress is a "plague of the 21st century". A healthy lifestyle, and the practices of yoga and meditation, which are various types of psychophysiological activity, are becoming a priority for people around the world [1]. In the world, the popularity of recreational psychophysiological exercises and psychological regulation is due to the need to resist the growing tension of everyday life, as well as the increasing desire of people to explore, protect, and expand their own spiritual potential [2,3]. At the same time, recent research studies studying the human brain have shown the positive effect of psychophysiological exercises on the human brain [4,5]. In other words, the practice of meditation can help a person cope with stress and depression. Scientists at Harvard University showed that after 8 weeks of meditation, the structure of the brain changes, and the density of gray matter in the

area of the brain responsible for memory and learning increases. It is worth noting that meditation practices, in fact, are aimed at focusing a person's mind on a particular object and using internal means of regulation.

Within this paper, we propose to solve a fundamental problem and aim at confirming the hypothesis that the external behavior of a person depends on his/her internal (mental) state. The paper is based on our previous work [6], where we considered dynamic motivational strategies related to the human-oriented daily life activities. State-of-the-art methods of artificial intelligence allow us to solve the problem of classifying human behavior based on the analysis of external signs obtained as a result of analyzing images from a smartphone's camera and making conclusions about his/her internal state. We propose to classify the meditation process using image recognition techniques, as well as using information from wearable electronic sensors to estimate the quality of the practice. The relevance of the problem is confirmed by the recently growing demand for knowledge regarding the meditation practice and the widespread dissemination of this practice among the masses (for example, using the Headspace mobile application, whose annual income is USD 100 million), which emphasizes the social significance of the research. The scientific significance of the work is confirmed by the lack of models and methods today that allow a person to get an objective assessment of the quality of the meditation process based on an analysis of psychophysiological indicators.

The methodology of the automated assessment of the human meditation practice, which is the fundamental objective of the project, is represented by three key processes. (1) First, there is the preliminary training of the system and classification of the meditation practice, which is based on a comparison of the data array formed from images obtained from the video camera and indicators read from motion sensors (accelerometer) and heart rate sensors within wearable electronics devices. Readings of electroencephalograms are obtained using specialized equipment and data from a questionnaire filled out by the person at the end of the practice of meditation in order to make a subjective assessment of its quality. (2) Next, there is a software-based automatic assessment of a person's meditation practice, using a smartphone's camera (aimed at the person during the practice) and wearable device sensors. (3) Lastly, there is the search for a meditation coach based on their competencies, employing user preferences to perform a deeper assessment of the meditation practice based on the history of a person's work in the system.

The rest of the paper is organized as follows. Related work on the topic of the heart and breathing rate's correlation with psychophysiological activity, as well as image recognition techniques for human activity detection, is presented in Section 2. We also consider research work related to dynamic motivation strategies, which can be applied to the task related to motivation of the human into everyday practice. Methods are presented in Sections 3–5. We propose a reference model of the psychophysiological activity detection system in Section 3. Section 4 considers the meditation estimation approach and acquired dataset description. We propose a user motivation model for psychophysiological activity in Section 5. The main results are presented in Section 6. The conclusion summarizes the paper and contains the main discussion, which is informed by the results.

## 2. Related Work

We reviewed the work done in three areas of research, constituting the basis of our study. Firstly, we researched relationships between heart rate and psychophysiological activity. We analyzed the papers related to the study of the meditation process and found out that pulse is the most important factor as an indicator of calmness. We also considered relationships between breath rate and psychophysiological activity. Then, we considered image recognition techniques for human detection. We researched image classification and action recognition approaches.

Finally, we considered the competence management and motivational strategic approaches. In this area, we considered gamification approaches and examples of gamification's usage, as well as approaches to finding a coach for some activities. We also considered the expert networks,

that is, the community of professionals in some topic where the competencies of the participants are understandable.

## 2.1. Relationships between Heart Rate, Breath and Psychophysiological Activity

Human pulse and breath are the main measurements that characterize calm. The authors of the paper [4] identified them as the main psychophysiological activity factors related to the psychophysiological activity process (see Table 1). An example of pulse changes has been presented in an experiment about the differences between Chi and Kundalini types of meditation [7]. Figure 1 shows the pulse changing after the meditation practice starting. The heart rate varied in a range of over 30–35 beats/min within 5 s in some of the subjects. It correlates with breath rate, but the most significant fact is pulse decreasing after the meditation starting.

**Table 1.** Heart rate measures during the Acem meditation technique [4].

| Variable | Rest | Meditation | Difference | *p*-Value |
|---|---|---|---|---|
| Respiration rate/min | 16.9 ± 1.9 | 16.3 ± 1.8 | 0.52 (−0.05 to 1.9) | 0.072 |
| Heart rate/min | 72.6 ± 10.9 | 71.6 ± 10.6 | 0.92 (−0.15 to 1.98) | 0.088 |

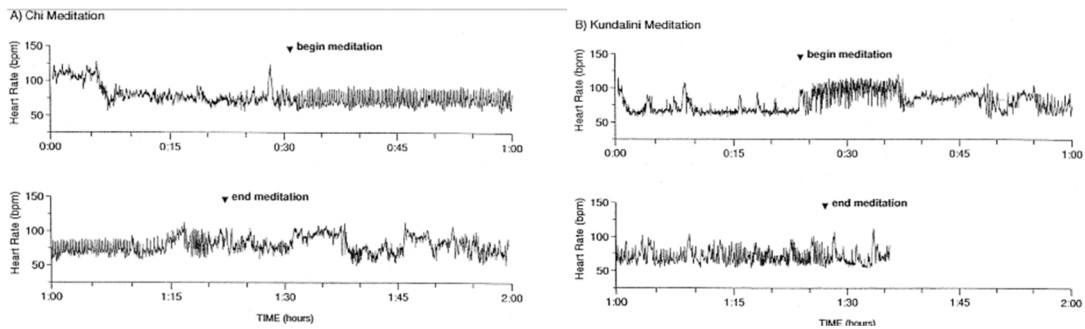

**Figure 1.** Heart rate changes during Chi and Kundalini meditation techniques [4].

Another experiment shows the increasing of the parasympathetic and reducing sympathetic nerve activity as well as increasing overall heart rate while practicing the Acem technique [8]. The presented heart rate is less than the average human heart rate. Two presented experiments show that pulse decreasing does not depend on the meditation technique. Moreover, this decreasing is also presented in general meditation exploring [9,10], but the decreasing tendency cannot be easily unified.

One more experiment has been studied in [11]. The authors say that different meditative/breathing protocols may evoke common heart rate effects, as well as specific responses. The results support the concept of a meditation paradox, since a variety of relaxation and meditative techniques may produce active rather than quiescent cardiac dynamics, which are associated with prominent low-frequency heart rate oscillations or increases in the mean resting heart rate. A similarity in heart rate variability during the relaxation response and segmented breathing can be noted. Both interventions induced prominent, relatively low-frequency oscillations in heart rate, which were associated with slow breathing at the same slow frequency. Compared to baseline, for both the relaxation response and segmented breathing, there is an increase in dynamic range, with an up to 20 beats/min change in heart rate occurring over 10 s. The respiration spectrum during segmented breathing shows a dominant peak at the mean breathing rate, as well as higher harmonics due to the nonlinearity of the signal.

In contrast, during the breath of fire, the respiratory rate is markedly increased (up to 132 breaths/min). Compare this response with the atypical pattern noted in one of the subjects, where breath of fire was associated with low-frequency heart rate oscillations.

Another considered body measurement is breathing. Visual analyses of the data showed a decrease in respiration rate during the meditation from a mean of 11 breaths/min for the pre- and 13 breaths/min

for the post-baseline to a mean of 5 breaths/min during the meditation, with a predominance of abdominal/diaphragmatic breathing. There was a noticeable decrease in respiration rate during the meditation (5 breaths/min) when compared to the pre- (11 breaths/min) and post-baseline periods (13 breaths/min).

The decreases in respiration from 11 (pre-baseline) and 13 (post-baseline) to 5 breaths/min, with continued slow breathing at 5 breaths/min during the meditation period, demonstrate significant meditative control (see Figure 2). It suggests a decrease in arousal that appears consistent with other findings on the physiological correlates of meditation that show a decrease in respiration. Although previous studies have reported the breathing rate decreasing, the pattern of breathing is not usually noted. In this subject, there was a consistent increase in abdominal over thoracic breathing seen during the meditation period. The abdominal/diaphragmatic breathing rate seems to be an integral part of many relaxation techniques [12].

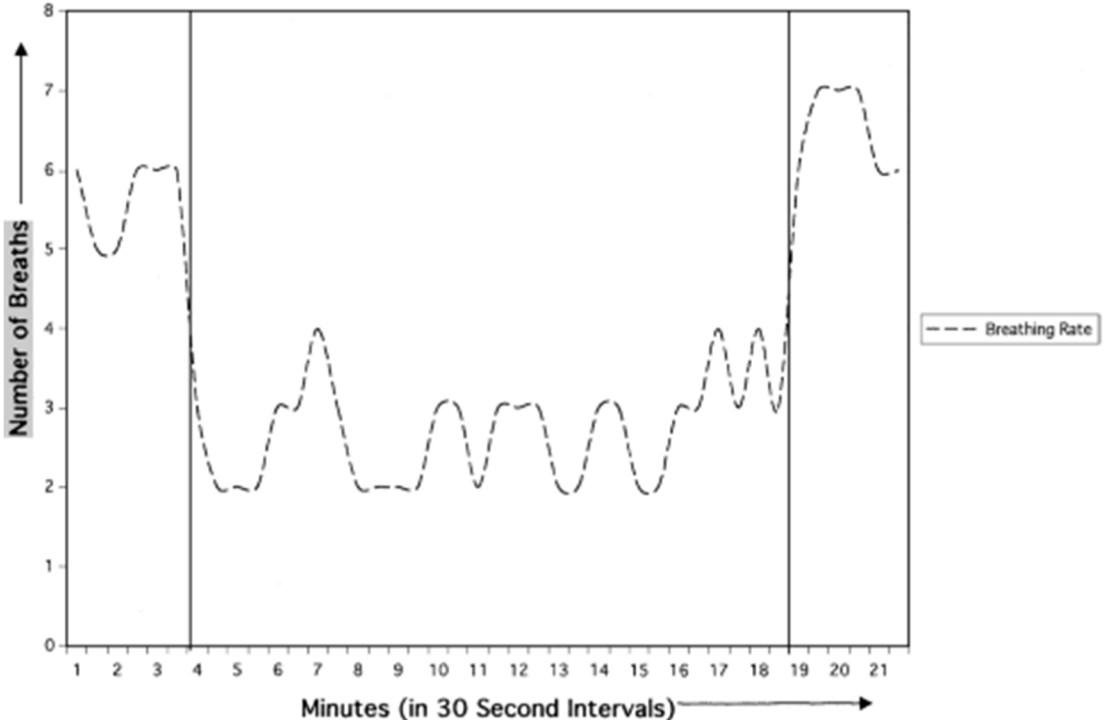

**Figure 2.** Breathing rate during the meditation (pre-, meditative, and post-baseline periods) [8].

Other researches also show the decreasing of breath rate [10] during the meditation process. Moreover, some of them prove that this decreasing is similar for beginners as well as for experienced people [13]. There are some other body measurements that can be correlated with the meditation process—for example, blood pressure. It is not certain whether it is truly superior to other meditation techniques in terms of blood pressure lowering because there are few head-to-head studies. However, meditation techniques in general do not appear to pose significant blood pressure changes [11,13]. However, the meditation process does not guarantee reducing blood pressure, but it also does not cause it to increase (see Figure 3).

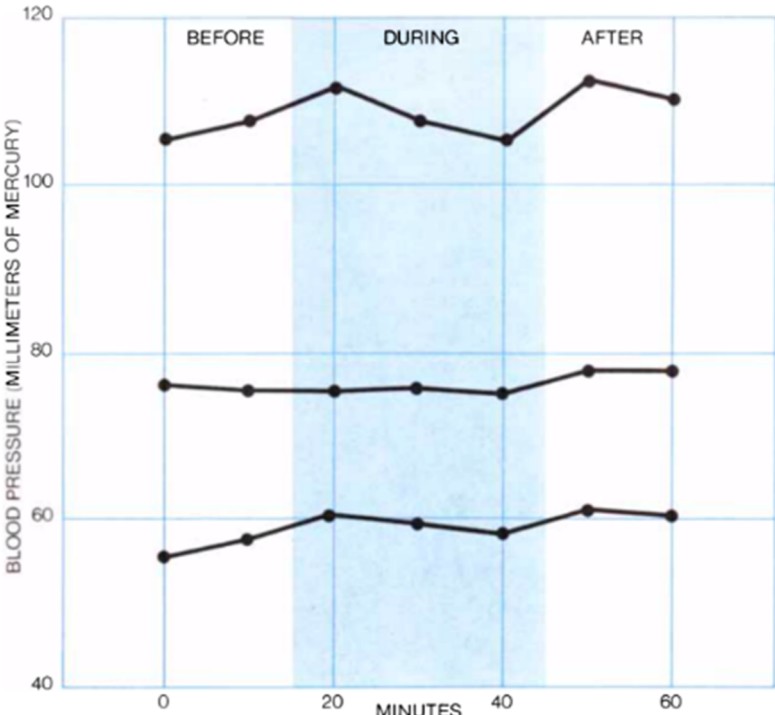

**Figure 3.** Blood pressure changes during meditation.

The generalization of considered research will be the conclusion that during the meditation process, a person should have a heat rate between 60 and 85 beats per min, and the upper bound for breathing rate should be 10 breaths per min.

### 2.2. Image Recognition Techniques for Human Activity Detection

Over the last few years, there has been a lot of research and development of methods for solving a wide range of recognition problems. Such problems include human action classification, determining position in space and time, and quality estimation of physical activity. We propose a neural network model to solve two main problems: video frames classification for pose recognition during the meditation practice as well as human behavior recognition and estimation.

We analyze different modern approaches used for solving similar tasks. For example, OpenPose [14] is one of the most popular bottom–up approaches for multi-person human pose estimation. Based on the key points of the human body, the system determines its basic position in space. Another example is the convolutional neural network for yoga pose classification [15]. It determines the pose taken by a person during yoga, including some postures for meditation.

The second problem is neural networks for action recognition. Several works discuss effective convolutional neural networks for action recognition [16]. Some papers include recurrent neural networks and long short-term memory [17]. The authors of the paper joined these two approaches into long-term recurrent convolutional networks [18], which is used for visual recognition and description. The authors of the paper [19] divide the network into temporal and spatial parts. In papers [20,21], the authors discuss skeleton feature matching sequences from images, improving the accuracy of defining the action and expanding the context of the action. However, the represented approaches have been tested on video data with significant and obvious movements such as walking, swimming, etc. None of them has been tested on a motionless process such as meditation.

Different human pose datasets exist on the Internet—for example, the MPII human pose dataset or dataset for CNN yoga pose classifier. To recognize the skeleton of the key points of a person, we use specialized datasets—for example, the COCO keypoints challenge dataset and VGG pose dataset. However, these sets should be expanded. The approach using synthetic data [22] can help with this.

In our work, we should be able to recognize the human behavior by video frames as well as give an estimation of his/her actions. The solution to basic actions was demonstrated by researchers in the paper [23]. However, human movements in the video are easily tracked and significant. In our case, the person is in a state of rest. This may complicate the assessment task. To determine the position of the body more accurately, researchers collect more key points of a person. For example, this approach even allowed simulating a character moving in time with a person from polygons based on key points [24]. In addition, we combine human skeleton data with three-dimensional temporal images, as presented in the paper [25].

*2.3. Competence Management and Motivational Strategy*

Psychophysiological activity, similar to almost any occupation, is much more effective to study under the supervision of a competent coach. The task of searching for a good coach for physical activity is important and actual for the considered domain. We investigate the coach networks that consist of communities of professionals in some topic where the competencies of the participants are understandable. There are no direct analogs for the system where psychophysiological activity coaches' profiles include their competencies. There are systems such as [26], where a person can find a meditation coach, but the competence of the coaches in such systems is not formalized.

Moving away from the topic of meditation itself, there are systems for selecting coaches [27] or systems for selecting tutors [28]. In such systems, some information about the competence of coaches is often present; however, these competencies are defined externally (such as through certificates). In some cases, the coach's competence is determined by the average of the ratings given by users, while there is no distribution of specific competencies. For the considered topic, the coach network brings together psychophysiological activity coaches and humans who want to learn psychophysiological practice. One of its main mechanisms is coach assessment.

One way of defining a coach's competency level in the system is estimated on the basis of deviation from the average assessment by other coaches. This approach is confirmed by the paper [29], where some students were surveyed on complex questions that did not have a clear answer. A survey of many students was more accurate than individual student responses.

An important task of the developing system is the competent coach selection for helping a user with certain characteristics (gender, age, etc.) in a meditation practice. The task of the coach finding for a given topic is considered in paper [30]. The authors propose a method for the coach assessing based on the contents of each coach's documents. This method was determined to be the most accurate relative to others.

The authors of the papers [31,32] prove that coach judgments are subject to a number of cognitive and motivational biases, as well as the concrete coach context and his/her personal experience. In the context of the approach presented in the paper, there is a danger of prejudice based on the appearance of meditation coaches. For this work, this led to the use of comparison with the automatic meditation assessment by a neural network.

The authors of the study [33] propose a model for changing the significance of coach estimates to improve probabilistic forecasts. Preliminary studies show that coach judgments are subject to several cognitive and motivational biases, the specific coach context, and his/her personal experience. The authors show that the coach competence should be assessed based on how accurately coaches answer the questions. To make the decision, the authors propose performing mathematical aggregation of coach estimates.

Another important issue related to the meditation practice is to motivate the human to implement the practice regularly. Gamification is the approach to work with a user's motivation [23]. For our information system, gamification will be used to motivate a user to devote more time to the meditation process. The term gamification is relatively new, and the usage of gamification in information systems is increasing. Systems are increasingly gamified, and the implementation of games elements in the real

world is common nowadays. This trend continues due to gamification elements helping people better immerse themselves in the information system by influencing certain psychological needs.

The paper [34] considers gamification as a tool for motivating specific behaviors with certain game elements. The authors claim that it is expected that gamification can foster the initiation or continuation of goal-directed behavior, i.e., motivation. There are connections between gamification elements and specific psychological user experience. Three basic psychological and intrinsic needs are postulated: the need for competence, the need for autonomy, and the need for social relatedness. Each of these needs has a connection with particular game elements. To one degree or another, each element of gamification satisfies one of the needs.

For the meditation motivation approach considered in this paper, these needs are closely related to the reasons why the human uses our system. The answer to the question: "what led the user to us?" helps select the necessary elements for better interest and immersion in the system.

### 2.4. Related Work Results and Task Definition

Examined studies proved the independence of changes in heart rate as well as respiratory rate during meditation, which was based on the type of meditation. We also determined the normal ranges for these indicators. The generalization of considered research allows concluding that during the meditation process, a person usually should have a heart rate between 60 and 85 beats per min and an upper bound for breathing rate of 10 breaths per min. We studied patterns of human behavior during psychophysical activity. In this paper, we check these patterns by analyzing the video stream and wearable electronics data.

We identify the following requirements to the system that estimate the human psychophysiological activity and motivate him/her to such activity in everyday life.

- Meditation evaluation support based on human video monitoring as well as heart and breath rate measurements.
- Intelligent search function support for coach search based on his/her competencies as well as human preferences.
- Automatic proactive audio guide proposal based on human experience.
- Dynamic motivation based on human preferences, including gamification elements.

## 3. Reference Model

The reference model represented in Figure 4 shows the main system components and relationships between them. Data for evaluation are based on tracking the person's body movements, heart rate, and breath rate collected from possible sources: a smartphone camera and wearable devices data. A smartphone can be used as a universal device that has a built-in camera for tracking the person's body as well as the possibility of connecting wearable electronics. Modern wearable electronics allow analyzing the human heart rate and movements during meditation that can be used for the process estimation. We identify the main patterns of human behavior based on heart rate and activities measurements. Based on this pattern, we build a wearable-device based evaluation of meditation.

By utilization of the meditation estimation model and computer vision techniques, we use video data to classify human behavior using a previously trained neural network. Therefore, based on this classification, the system gives an estimation score of the process. We propose in the paper the process of neural network training based on a meditation dataset that allows recognizing the pose of the human during the meditation process. Based on video data, we build an image-based evaluation of meditation.

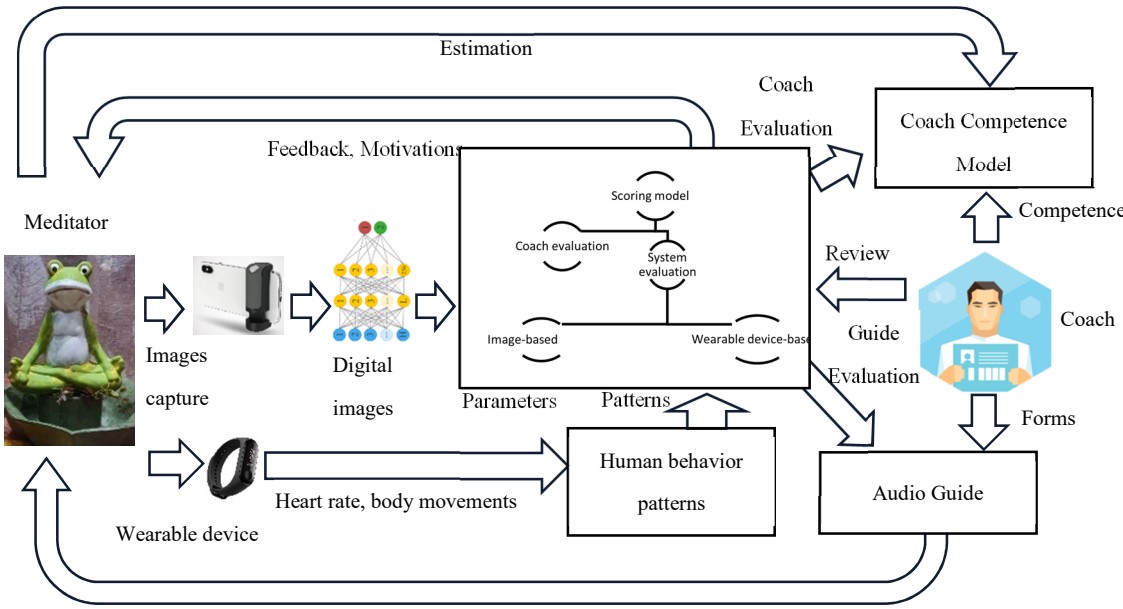

**Figure 4.** Reference model of the psychophysiological activity detection system.

A coach could review the meditation. An overall meditation evaluation consists of a coach evaluation merged with image-based and wearable device-based ones. The overall meditation evaluation influences the audio guide selection: the user receives an audio guide that gives them the best chance of meditating better. Based on the user's meditation evaluations, the motivation model is also built. Motivations and coach feedback are shown to the user. If the user was supported by a coach, he/she could give his coach a rating for mentoring. This rating of the coach's competence influences their selection in the future.

## 4. Meditation Estimation Approach

### 4.1. Patterns of Human Behavior Based on Wearable Electronics

We determine four main patterns of human behavior that are related to the meditation process. Patterns require continuous measurement of the heart rate and acceleration parameters of the human body. The most affordable tools of measurement in this case are wearable electronics devices, which have risen in popularity in the last few years. We made measurements for verifying distinguished patterns. In the experiment, recipients performed various body movements, experienced emotions, and meditated. The measurements involved completely healthy people. Recipients did not have any diseases associated with the heart or vascular system. The main parameter was the heart rate and movements.

The first pattern is a completely calm pattern. In the measurement for this pattern, a recipient was in a state of calm during meditation, and they were not affected by external and internal factors (see Figure 5). The linearity of the heartbeat and the minimum level of acceleration characterize the calmness of a human. However, not all recipients were able to achieve complete calm. The heart rate of Person 1 jumped briefly, which can be caused by a short exciting thought.

Emotional patterns are related to significant increases in heart rate while human activity is minimal. Figure 6 shows that the heart rate depends also on the emotional state of a person: on his or her thoughts. The heart rate of some recipients increased and decreased smoothly, while the heart rates of others changed sharply. This feature is caused by the internal characteristics of an individual's thinking, which influence his/her body and will be taken into account in further studies.

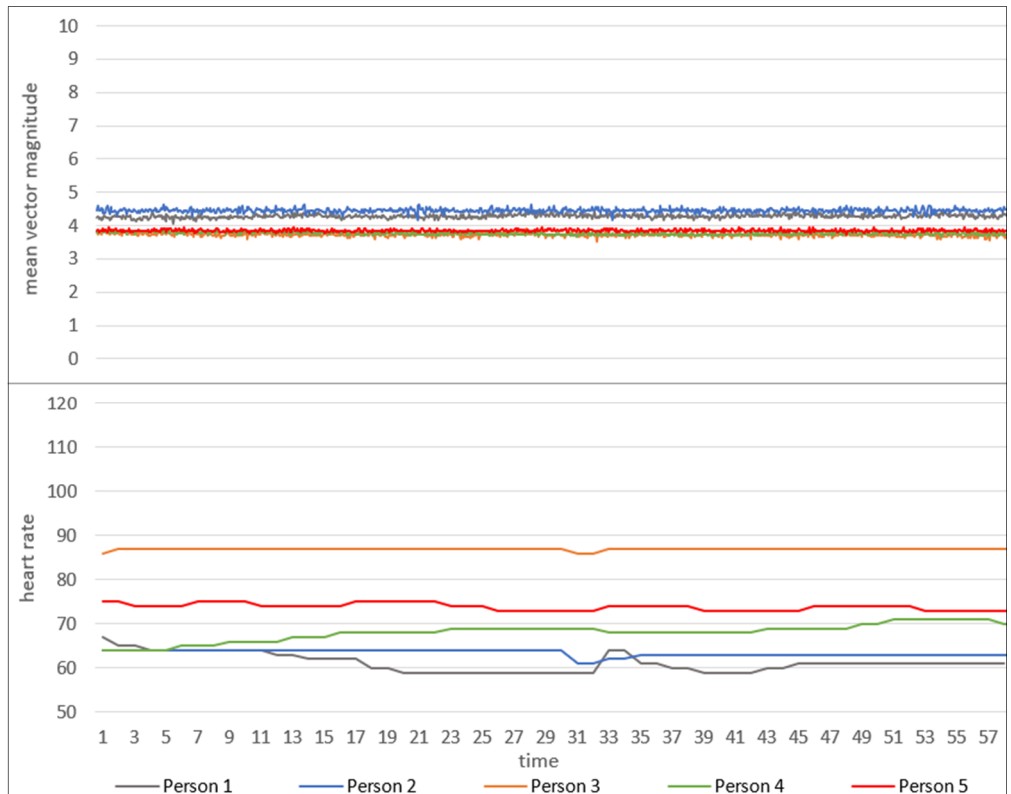

**Figure 5.** Heart rate and acceleration for the complete calm pattern estimated for five people.

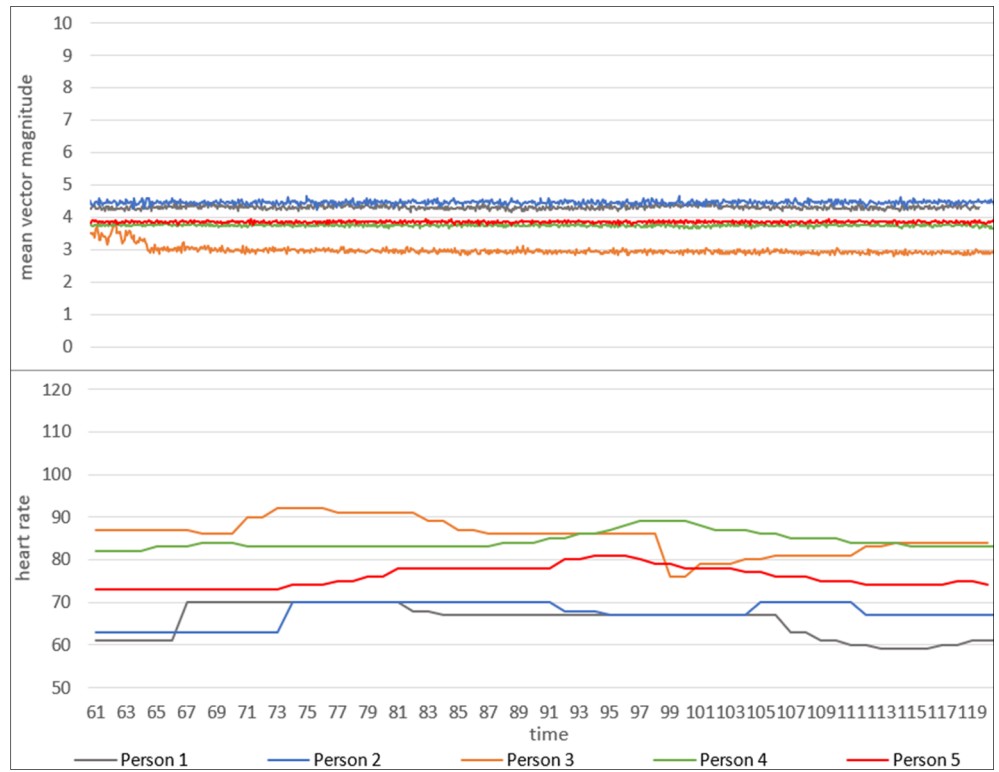

**Figure 6.** Heart rate and acceleration for the emotional pattern.

The next considered pattern is a gesture pattern. In the measurement for this pattern, we asked a human to make some gestures. Gesture patterns are characterized by small body movements such as scratching or small movements of the hands. In these instances, the heart rate of the person is not increased. At the same time, the level of activity is spasmodic (Figure 7). Small gestures practically do not affect the heart rate. However, the heart rate of People 1, 2, and 3 was increasing. Probably, for some people, even small body movements cause a strain on the heart. Large single movements of body parts have a minor effect on the heart rate. During small body movements such as scratching or small movements of the hands, the heart rate of the recipients did not increase. At the same time, the level of activity was spasmodic.

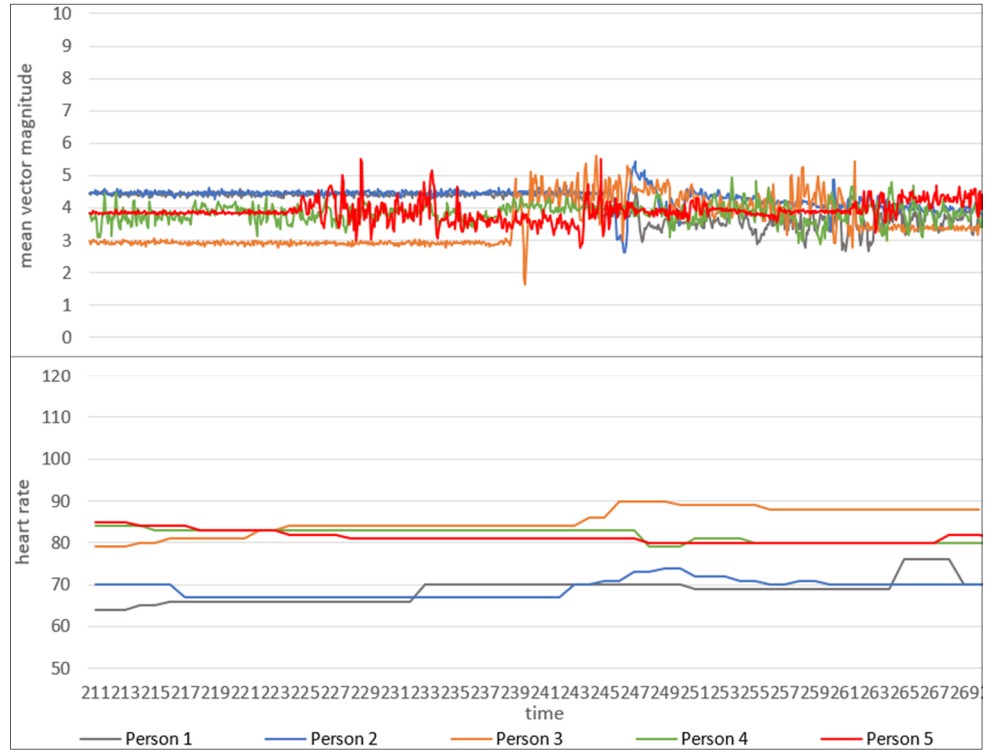

**Figure 7.** Heart rate and acceleration for the gesture pattern.

At the end, let us consider the situation in which a person moves during the meditation practice. During the movement pattern, the heart rate of a person rises in accordance with the level of physical activity when a person performs his/her daily activity or various physical exercises (see Figure 8). It is obvious that during meditation, a person should be in a practically motionless pose. This will allow him/her to focus on the process and achieve complete peace of mind. The heart rate of some people was increasing, while the hear trate of others was jumping during the movement. Different heartbeat behaviors are caused by different physical parameters of recipients and their activities.

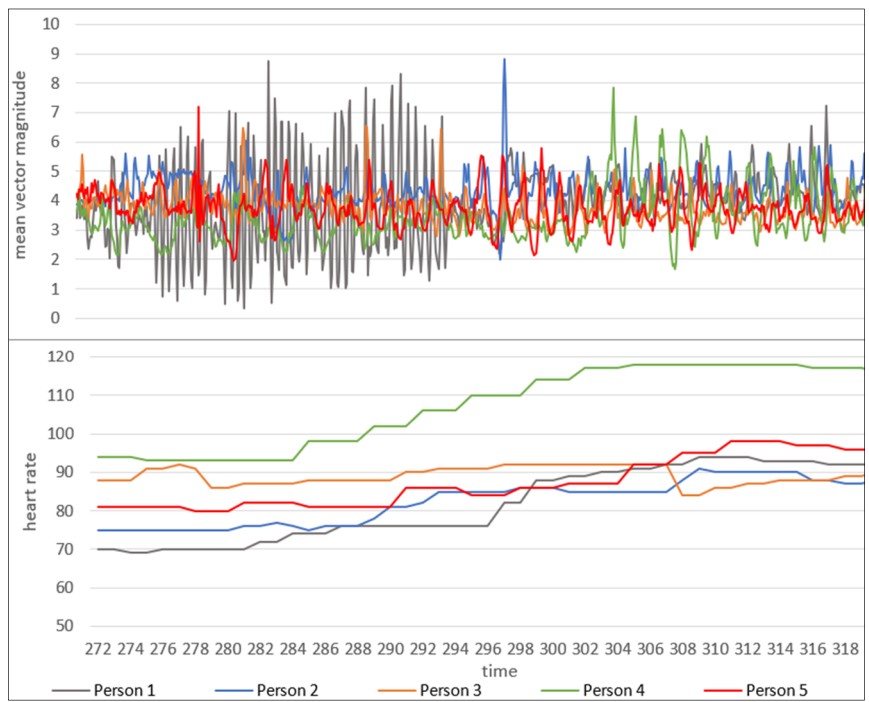

**Figure 8.** Heart rate and acceleration for the movements pattern.

## 4.2. Meditation Video Dataset

We constructed the Meditation Video Dataset (https://cais.iias.spb.su/meditation/index.html), which contains videos of meditating people tracked by a smartphone camera. The dataset contains more than 70 videos of 17 people. Each video is around 20 min. We divided meditators by two levels of experience: beginner and professional. We collected these data manually from friends and students who practiced meditation by recording videos as well as several professional meditators. The dataset consists of 8 meditations from professional meditators and more than 50 beginner's meditations. It includes 20 meditations done by women and more than 50 by men. The vast majority of videos were recorded in two poses: lotus-based (see Figure 9) and sitting on a chair (Figure 10). Professional meditators chose the lotus pose while beginners chose the lotus-based pose as well sitting on a chair. We also collected feedback from meditators, which they recorded after meditation. We ask mediators to include in feedback their experience and a mark of how well they practiced meditation on a 10-point scale (Figure 11).

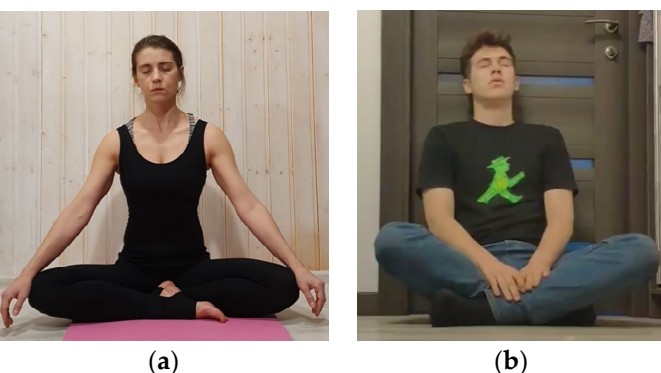

(**a**)　　　　　　　　　　　　　　　　　　(**b**)

**Figure 9.** Dataset example: (**a**) professional in lotus pose, (**b**) beginner in lotus pose.

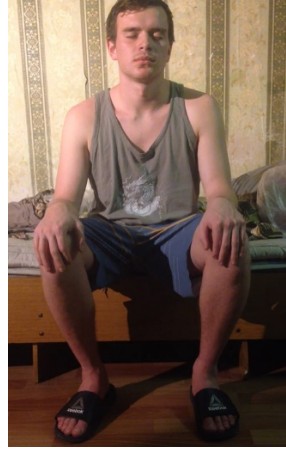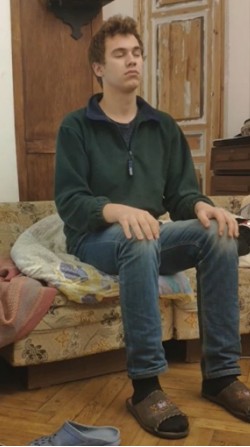

**Figure 10.** Dataset example: beginner in sitting position.

1. It was easier than the first time (the video for the first meditation was not recorded), I sat down comfortably, but I got the feeling that the brain was working and the body was falling asleep, possibly because of fatigue. Toward the end, there was a feeling that 30 minutes had passed, which even looked at the time. It was possible to concentrate better and be less distracted by extraneous sounds.
Rating 5/10

2. Time did not last as long as last time, but it was possible to concentrate worse than before. Towards the end, I got the feeling that I couldn't sit already in one place, tried to calm down, then it became easier, I managed to concentrate, but the alarm rang.
Rating: 4/10

3. I decided to try to meditate less time, because in the end it still becomes difficult. This time, everything went well, there were no superfluous thoughts, and I was able to calmly concentrate and even managed to imagine that I was in a different place.
Rating: 7/10

4. I couldn't wait for an alarm this time, I couldn't get together at all. I meditated between work, maybe because of this I could not relax, because they could write at any time, and there were thoughts about work tasks
Rating: 3/10

5. Increased time, about half went well, then it became uncomfortable to sit again, after it turned out to relax and concentrate, that even when the alarm clock rang, I started.
Rating: 5/10

**Figure 11.** Dataset example: Meditations feedback by one of meditators.

*4.3. Meditation Estimation Based on Neural Networks*

We used the meditation dataset described in Section 4.2 for learning the neural network model to determine each individual's pose during the meditation practice. The model allows automatically detecting if the person sits in the right pose and if he/she changed it during the process (see Figure 12). In order to recognize the correct pose, the neural network has to receive a normalized array, which is essentially a black and white image. With the aim of recognizing, the meditation process an array of consecutive pictures is fed to another neural network. The response of this neural network is an evaluation of a certain period of the meditation. After evaluating the meditation, these estimations are summarized, analyzed, and converted into the final rating that the user sees. We use the following technology stack to create the neural network model and work with the video data:

- TensorFlow Machine Learning Library 2.1,
- Keras 2.3 as a high-level neural network API for TensorFlow and another machine learning library,
- OpenCV computer vision library 3.4.
- Python 3.7 due to the large community and high support of TensorFlow.

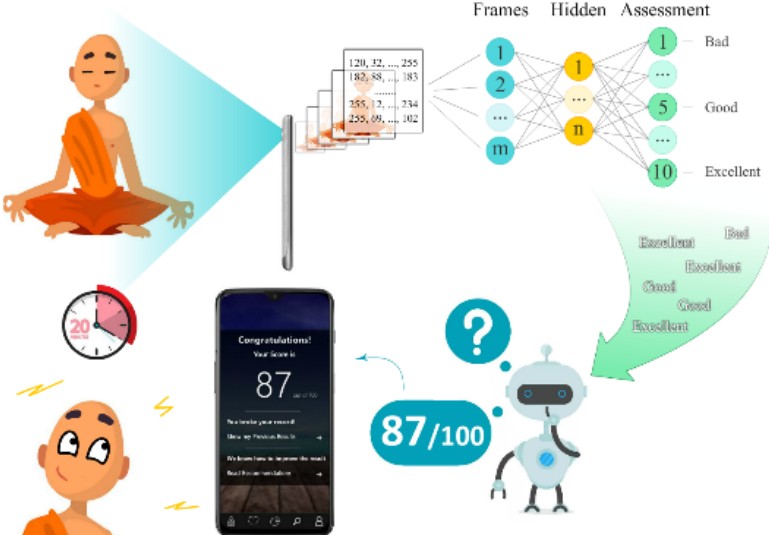

**Figure 12.** Neural network-based meditator seating pose detection.

Firstly, we learned the neural network that aims to recognize the correct pose of each human in the meditation process. We used the dataset described in Section 4.2. Meditation pose recognition is a classification task. To solve this task, we proposed using a convolutional neural network (see Figure 13).

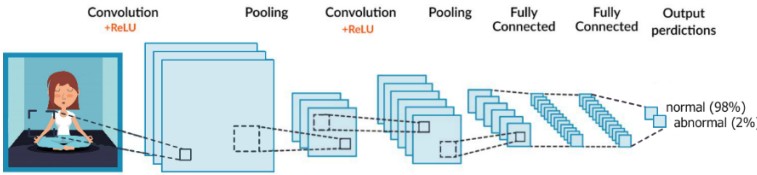

**Figure 13.** Convolutional neural network for meditation pose estimation.

The neural network consists of several consecutive convolution and pooling layers. This approach allows us to classify images more accurately. The proposed convolutional neural network consists of three consecutive pairs of such layers and one more layer, called ZerroPadding2D. This layer adds black dots to the borders of the image, since the convolution layer removes these borders.

An important task is training the neural network model; it is necessary to prepare and qualitatively label the dataset. For this task, we develop an algorithm that converts all video used for the learning stage into a set of frames (see Figure 14).

Then, these images are labeled and divided into two classes: the pose of meditation and the other pose. The program that is responsible for training the neural network collects all the images and inputs them into the model for training (see Figure 15).

A common approach to the classification model training is to modify images before feeding them to the training dataset. The training process takes place in several stages, which are named epochs. Each epoch consists of the same dataset. We transform every image before each step; then, training will be slower but more efficient. We apply slight scaling, rotation, and shifting of the images.

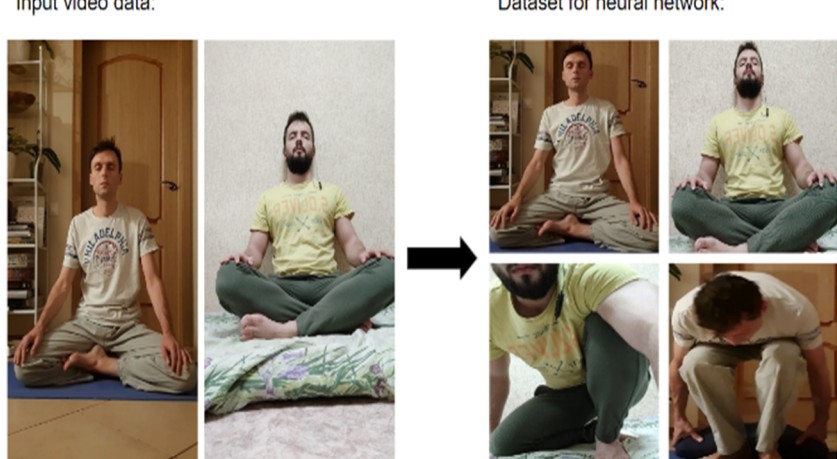

**Figure 14.** Converting video data to $180 \times 180$ px pictures.

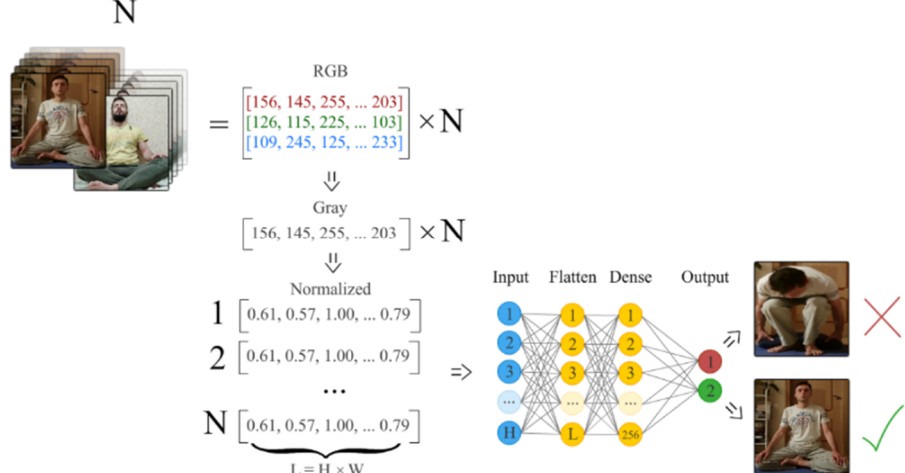

**Figure 15.** Image normalization before training the neural network.

Thus, after the transformation step, the neural network gives the results (see Figure 16). This figure shows two graphs (a and b). Graph (a) shows the accuracy of the model prediction. Graph (b) shows the value of the loss function. The result for a training dataset is shown by the blue line, and the result for the test set is shown by the yellow line. So, the model is trained well based on the first 10 epochs, so we stop the training process to avoid overfitting.

We increase the training dataset to get higher accuracy. The result is shown in Figure 17. A peak in the predictions' accuracy (see graph a) of the obtained neural network is achieved at the 10th epoch and amounts to more than 89%. This is enough at the current stage of work. Graph (b) shows the loss function. Figure 18 shows two screenshots of the meditation pose estimation example. The pose parameter shows the value of the prediction accuracy for the current pose. The AvgPose parameter shows the average value of all previous measurements during the meditation.

Further improvement of the pose recognition model for meditation comes down to collecting new data and continuing to train the neural network. In particular, synthetic data are an excellent example of the quality of the data that can be used, which are collected by modeling images on a computer. The image of a meditation pose modeled on a computer is easy to modify; thus, it is also easy get a large set of labeled data. This can simplify the process of finding human pose images.

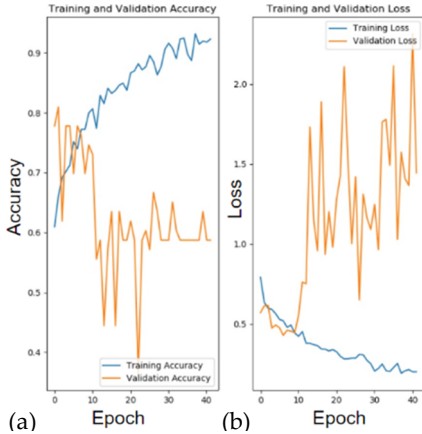

**Figure 16.** First results of a neural network. Graph (**a**) shows the accuracy of the model prediction. Graph (**b**) shows the value of the loss function.

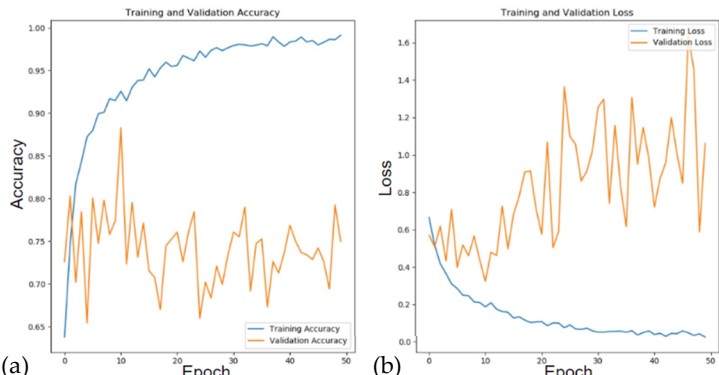

**Figure 17.** Peak accuracy of neural network predictions (**a**) and loss function (**b**).

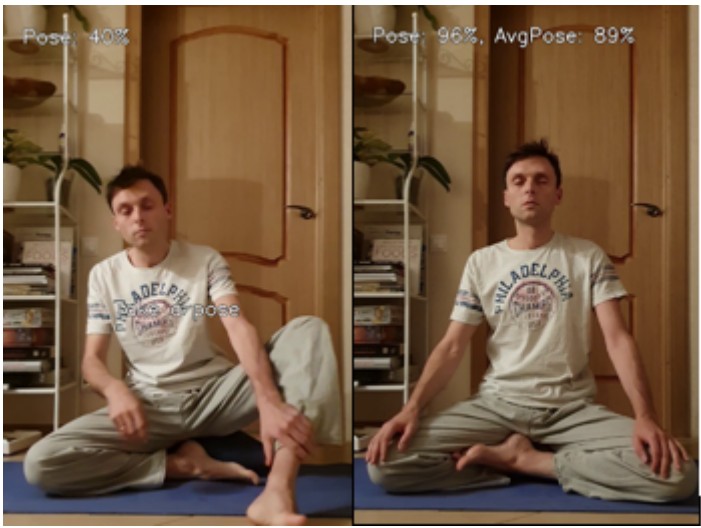

**Figure 18.** Examples of meditation pose recognition based on a developed model.

*4.4. Meditation Estimation Based on Skeleton Detection*

We propose an approach to analyze the meditation process that involves an optical flow utilization to detect the tiniest movement of individual parts of the human body marked with key points on the human skeleton. Optical flow is a vector field showing the movement of each pixel between

two frames. We evaluate the frequency and amplitude of respiration, stoop over time, displacement, and body vibration from side to side by analyzing the optical flow. Furthermore, this information can be used to assess the quality of the meditation technique and identify common mistakes.

We chose a section of the meditation video (see Section 4.2) where a person fixes their meditation pose. If a person changes the pose, we implement the new video section analysis. For the first frame of the appropriate video section, the human bounding box is detecting with a convolutional neural network [35], which is based on Mask-RCNN [36] and showed one of the best results on the COCO Detection 2017 (47.7 mAP) and COCO Segmentation 2017 (41.7 mAP) datasets [37]. Then, in the bounding box, the key points of the human body are evaluated using a separate neural network [38], which took second place in the competition associated with the Human 3.6 M dataset [39]. This neural network builds a skeleton of a human and marks key points on the intersection of human body parts. The following key points of the upper part of the body are taken for further analysis: nose, thorax, mid-torso (torso), shoulders, elbows, and wrists (see Figure 19).

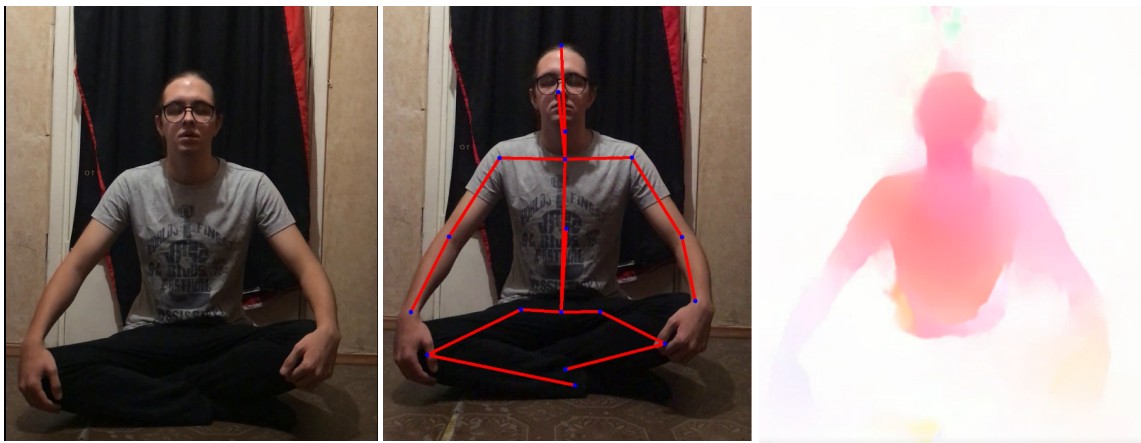

**Figure 19.** Human movement estimation based on skeleton detection.

Then, between the first frame of the video section and all subsequent ones, the optical flow (pixel displacement vectors) is calculated. For this purpose, a separate neural network is used [40], which showed one of the best results on the MPI Sintel [41] dataset (2.040 EPE matched). By calculating the optical flow relative to the first frame, a shift in the human body from the start of meditation is obtained. The graph constructed on the basis of such an optical flow is better visually perceived than the graph of the displacement relative to the previous frame, and the detection of key points of the body is necessary only for the first frame. Potentially, though, the bias can be large (for example, if a person is very stooped); this can affect the accuracy of some algorithms, including the widespread Farneback algorithm [42], but the neural network-based approach [40] that we used is resistant to strong displacements. Such a graph can also be obtained by summing the displacement vectors between pairs of adjacent frames, but summing the individual optical fluxes can lead to an accumulation of error, and it would also require a recalculation of the position of the key points.

Then, after calculating the optical flow and the position of the key points of the human skeleton, three motion graphs are formed for each key point and the whole body. These graphs show the displacement of the key point (or the whole body) along the X axis (left/right), the Y axis (up/down), and the absolute length of the displacement vector (takes into account both components). In Y, movement during breathing is better observed; in X, extraneous unwanted body vibrations are detected. Absolute undesirable moving of key points over time can be detected. As a result, 30 charts were received (see Figure 20).

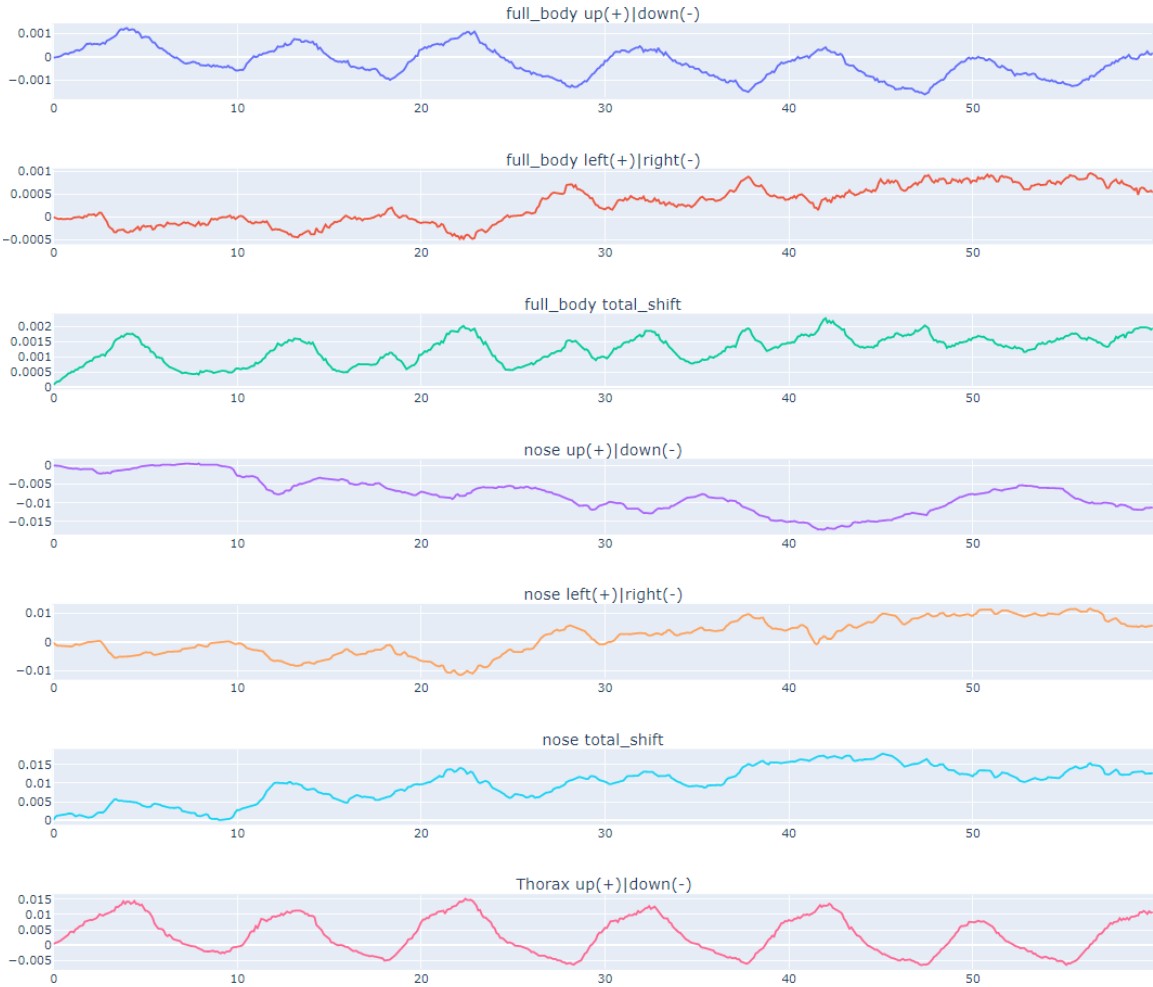

**Figure 20.** Human movement estimation based on skeleton detection.

For motion graphs formation of the whole body, the displacement vectors of all the pixels included in the bounding box of a person are averaged. To generate key point graphs, a weighted sum of pixel vectors included in the bounding box is also calculated, but the weights in this case decrease with distance from the corresponding key point. In the prototype, weights are calculated using the Gaussian function (see Equation (1)), where $x$ is the Euclidean distance between the pixel of the corresponding key point and the remaining pixels, $b$ is always equal to zero, $c$ is a parameter, and $a = \frac{1}{(c\sqrt{2\pi})}$. The parameter $c$ is calculated as a percentage of the distance between the key points of the shoulders in pixels to take into account the different size of the person's body, the distance from the camera, and various characteristics of the cameras. For this research, 5% of the distance between the shoulders was used as a value of parameter $c$. It should be noted that the camera should be located directly in front of the person.

The resulting displacements of the body and individual key points are measured in pixels and then divided by the distance between the key points of the shoulders to also take into account the above variations.

$$g(x) = ae^{-\frac{(x-b)^2}{2c^2}} \tag{1}$$

As a result of the analysis of the graphs obtained from the videos of meditation, the following observations were made:

- Oscillations caused by breathing are noticeable on most graphs that take into account vertical movement, but the amplitude and frequency are best reflected on the graph of movement of the thorax key point along the Y axis;
- The movements of the head are well reflected in the graph of the nose, but it also reflects stoop and other movements of the upper part of the body. To get only the movement of the head, it is needed to subtract the schedule of movement of the shoulders;
- The stoop is well observed on the shoulder graphs.

Using this technology, a real-time feedback for a coach can be organized. The coach can analyze the movement graphs of parts of the human body, draw attention to problems, recommend ways to solve them, and over time, check whether they helped. For example, the coach can recommend exercises to strengthen the back muscles, or relaxation techniques (various music, etc.) that will be individually selected. Errors can also be classified automatically, and recommendations for resolving them are provided. In addition, the person himself may be interested in observing his/her progress over time.

*4.5. Competence-Based Model for Meditation Coach Search for Practice Estimation*

We develop a reference model for a meditation coach's competence management. The model helps select the most relevant coaches for the human based on his/her requirements, preferences, and coach expertise. Different coaches are professionals in various fields and have different competencies. Moreover, we found out that for many users, the internal response to the coach is important. Psychological compatibility plays an important role. We rank coaches for each user based on the following three things.

- Reviews of the coach by his students with similar tastes and preferences.
- Accuracy of coach's assessment: how much the coach's grades of meditation are different from those of other coaches.
- Quality of audio guides recorded by the coach. How strong these audio guides helped people with similar preferences meditate better, which is determined by how the users rated the audio guides.

We check whether a meditation audio guide could assist a specific person to meditate better. If an audio guide really helps, we recommend this meditation audio guide to users with similar characteristics (age group, meditation aim, etc.). We will rank audio guides based on the following.

- Marks set by humans who used this meditation.
- Influence on meditation quality: how much the audio guides improve the overall meditation rating of the person's meditation.

Figure 21 shows the reference model of the main entities in developing mechanisms. We divide the developed mechanisms into three subsystems. The Audio Guide Analysis Subsystem includes a meditation audio guide module and audio guide rating mechanisms. The Meditation-based Recommendation Subsystem includes User, Meditation, and Recommendation model modules. This subsystem processes information about meditation and user characteristics and includes mechanisms for selecting a trainer and audio guide. The Expert Analysis Subsystem includes Review and Coach modules. This subsystem tracks user interactions with the coach.

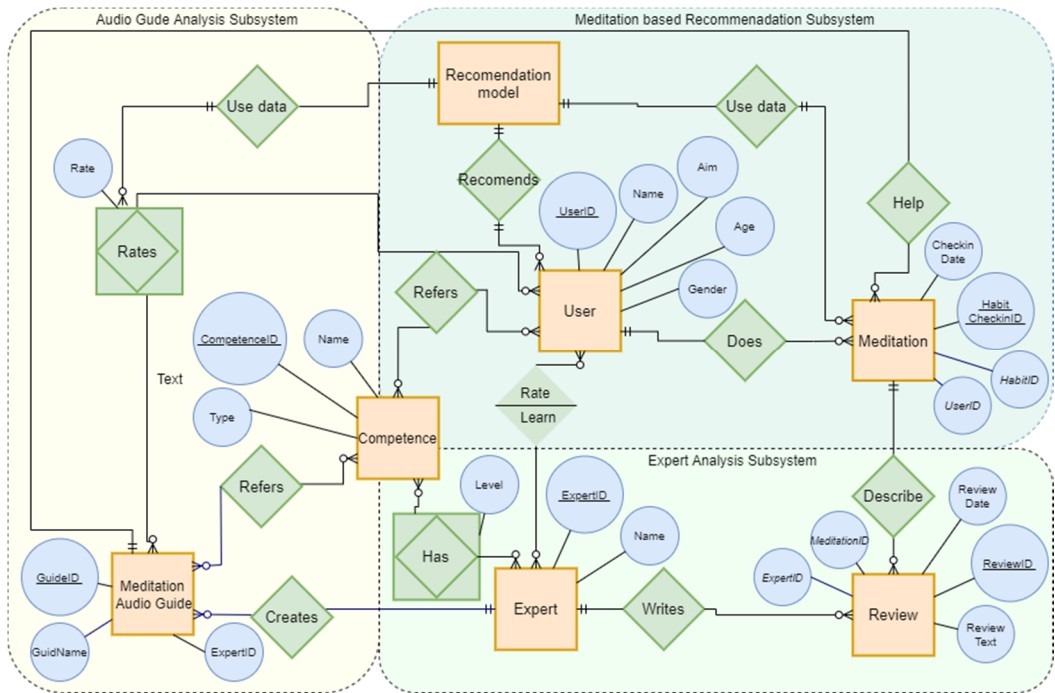

**Figure 21.** Reference model for competence management of conscious exercise coaches.

The evaluations of the audio guides are calculated based on the degree to which meditation with this audio guide is better than the average user's meditation score. The evaluations of the audio guides also affect the rated competencies of the coaches in relation to users of this type, so the specifics of the use of the audio guides influence the coaches' competencies levels. Levels of competencies are also calculating based on the accuracy of coach evaluations in the review of meditation as well as mentor evaluations by user. The competencies of coaches should be based on the following features.

- Goals of meditation: reduce stress, improve productivity, etc.
- Characteristic of user: age, gender, etc.
- Quality of user's meditations: beginner, advanced, etc.

## 5. User Motivation Model for Psychophysiological Activity

We develop the motivational model in such a way that includes three major elements: gamification elements, additional information, and social chats. These elements form 4 different motivational sub-models that are aimed at maintaining a user's activity in our application.

The presented sub-models and corresponding motivational elements are displayed in Figure 22. The basic motivational sub-model consists of general meditation resources and basic gamification elements. This sub-model allows interacting with and guiding the user through the application. Meditation resources selection by the user as well as interactions with the gamification elements helps the system understand which sub-model is more related to the user. We define the following sub-models.

The 'Learn to meditate' sub-model is focused on the user's progress in their meditation practice. Gamification elements such as achievements, the self-competition system, performance graphs, and the interactive meditation bot helps the user feel like they are making progress in the meditation and feel positive after each practice.

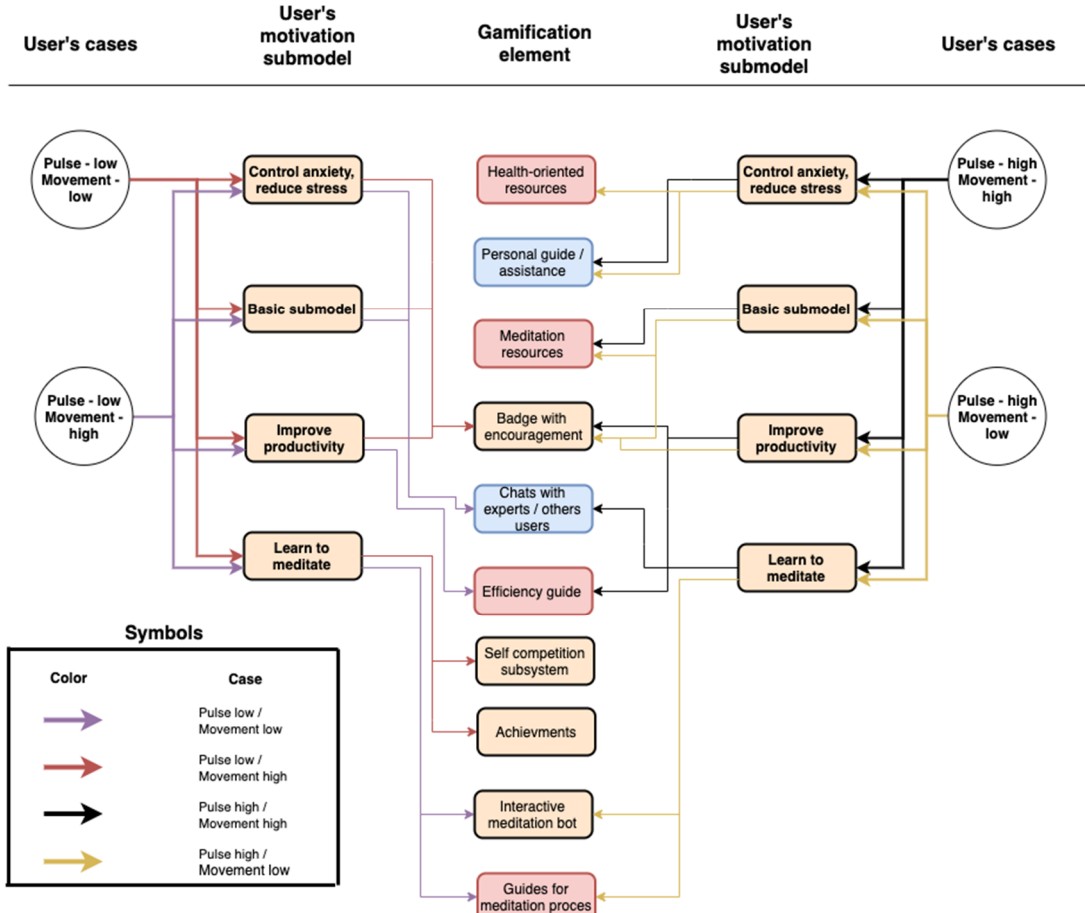

**Figure 22.** Gamification elements based on a user's motivational model and the results of their meditation practice.

The 'Reduce stress and control anxiety' sub-model helps the user overcome psychological problems such as stress and anxiety. Health-oriented resources consist of materials and courses selected by experts that can help achieve the goals of this sub-model. Gamification elements such as badges with encouragement and achievements show the user his/her progress with meditation practices. Communication with coaches and other users on psychological-related themes help the system assume that we can help users overcome their psychological issues.

The 'Improve productivity' sub-model helps the user improve his/her productivity. Gamification elements such as badges with encouragement, achievements, and performance graphs indicate the user's progress over time. The efficiency guides in the additional information section help the user stay focused on efficient time allocation.

Additional information selection, social chats, and user interests in configurable information such as performance graphs and achievements inside the application add points to each particular sub-model.

## 6. Evaluation

### 6.1. Meditation Estimation Evaluation

The section describes the developed meditation estimation model based on manual analysis of the dataset videos discussed in Section 4.2. Based on video analysis, we propose and evaluate the model for meditation process estimation. The aim of the manual video analysis is to determine parameters that are important for meditation process estimation. Based on these parameters, we propose the model that can be used for scoring the meditation process, which is shown in Table 2. The exact values of the

features are not so important. The most significant is their relationship to one another, which show what is more important and what is less. All of the meditations can be divided into three phases:

- Introduction phase: the first 2 min of the whole meditation time;
- Conclusive phase: the last 1–2 min of the whole meditation time;
- Main phase: the residual time period.

**Table 2.** Defined features and their grades.

| Parameter | Introduction Phase | Main Phase | Conclusive Phase |
|---|---|---|---|
| Pulse: The first decrease of more than 10% | +10 | 0 | 0 |
| Pulse: Increase of 10% from the previous measure | −5 | −15 | −10 |
| Pulse: Staying lower than 90% from the first measure | +5 | +15 | +10 |
| Breath: The first decrease for less than 10 breaths per min | +20 | 0 | 0 |
| Breath: Increase for more than 10 breaths per min | −5 | −15 | −10 |
| Breath: Staying lower than 10 breaths per min | +10 | +10 | +10 |
| Squaring shoulders | −3 | −5 | 0 |
| Straightening a back | −3 | −5 | 0 |
| Head movements | −3 | −5 | 0 |
| Changing the position of lower body | −3 | −10 | −5 |
| Changing the position of hands | −3 | −5 | 0 |
| Opening/closing eyes | −2 | −5 | 0 |
| Preserving the body position | +10 | +20 | +15 |

This means that features grades have to be also correlated with the meditation phase. We prepare the detailed description of several meditations from the dataset and analyze it to find correlations and tendencies during the process. Table 3 contains a description example of the meditation process done by a professional meditator. The process has been done using a lotus meditation pose, and the meditator takes the pose quickly. Meditation has been performed with tantric music accompaniment. As it can be seen from the table, the professional meditator performs the process without any interruptions, pose changes, or eye openings. Her breathing rate is quite small: 6–7 breaths per minute.

**Table 3.** Example of meditation analysis process done by a professional.

| Basic Aspect | Value | Comment |
|---|---|---|
| Meditation process duration | 22 min | |
| Amount of pose changes | 0 | |
| Amount of interruptions | 0 | No interruptions |
| Amount of eye openings | 0 | |
| Breathing rate | 6–7 | Deep breath |
| Breath type: thoracic/abdominal/other/not clear | thoracic | No changes |

Oppositely, meditations done by beginners look completely different. Table 4 shows an example. We conclude that the beginner meditator has a lot of small movement during the process, which is described in the table for the parameter amount of pose changes. We have analyzed a total of 8 meditation videos. This analysis shows that beginners also have interruptions during the process as well as eye opening.

**Table 4.** Example of meditation analysis process done by a beginner.

| Basic Aspect | Value | Comment |
|---|---|---|
| Duration | 15 min | |
| Amount of pose changes | Undefined amount | From 0:00 to 02:40 min<br>1—shaking, head up, and down movements;<br>2—sliding hands over knees;<br>3—swaying (impulsively);<br>4—lowered hands lower, raised head;<br>5—lowered head;<br>After 2:40 min:<br>03:00 min—raised and lowered head;<br>03:30 min—straightened shoulders,<br>straightened up, raised hands, raised head;<br>04:18 min—moved hands;<br>05:31 min—lowered hands, raised head;<br>05:48 min—raised head;<br>08:10 min—took a deep breath and<br>straightened up;<br>08:30 min—raised arms, straightened,<br>straightened shoulders;<br>09:45 min—lowered hands, straightened,<br>straightened shoulders, and raised head;<br>09:54 min—lower lip twitched;<br>10:12 min—"chewed" lips;<br>11:24 min—strongly lowered his head;<br>12:54 min—leveled head;<br>13:20 min—raised hands, straightened,<br>straightened shoulders, and raised head;<br>14:58 min—moved hands, straightened,<br>straightened shoulders, raised head, and took<br>a deep breath;<br>At the end, active head movements up/down<br>and lip biting. |
| Interruptions | 0 | |
| Eye openings | 0 | |
| Breathing rate | 5 | |
| Breath type | thoracic | No changes |

## 6.2. Motivational Model Evaluation

This section describes the connection between the motivational model and a user's wearable electronics data from Section 4. After a meditation practice, the user smartphone sends the results to the database. The analysis module distributes the results of meditation into four categories in the "Session" table in the database and assigns these categories to the user meditation session in the database. We developed the gamification element selection scheme based on wearable electronics data (see Figure 23). We also developed the following data structure (see Figure 24); we suggested keeping every practice in the "Session" table; more detailed data of each practice are stored in the "Sensor_Data" table.

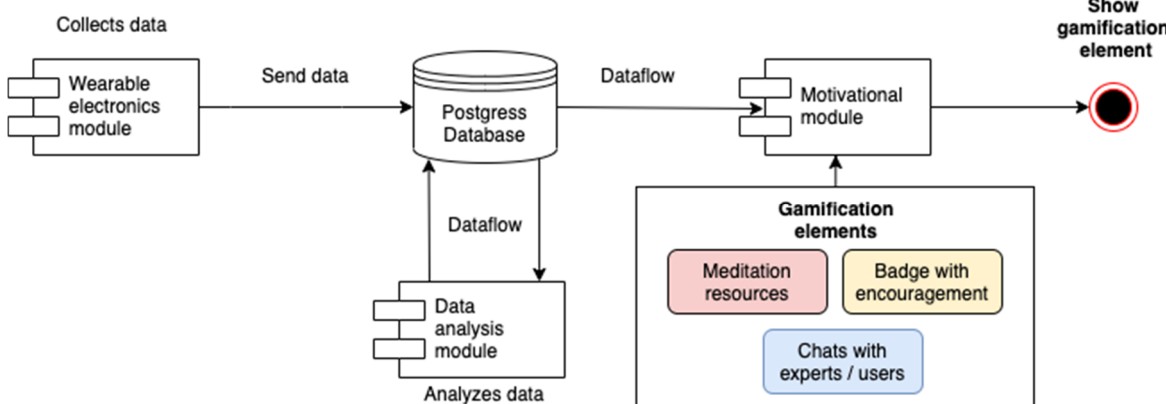

**Figure 23.** Gamification element selection based on wearable electronics data.

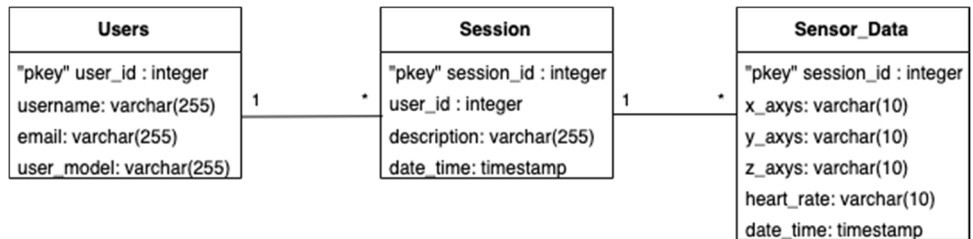

**Figure 24.** Database structure for meditation practices.

We present the following 4 patterns (sec Section 4) based on data from the user's wearable electronics to select the gamification element:

- Pulse low, movement low (see Figure 5);
- Pulse high, movement low (see Figure 6);
- Pulse low, movement high (see Figure 7);
- Pulse high, movement high (see Figure 8).

The data analysis module analyzes data from the "Sensor_data" table. This module transfer analysis results in the "Session" table in the form of the "description" cell. The description contains one of the four patterns named. Based on the results of the meditation practice and the user's motivational model (see Section 5), we assign a gamification element. To select a gamification element, the system checks the user motivation sub-model from the "Users" table and session description from the "Session" table. Based on these data, the system selects one or more gamification elements for displaying after the meditation practice. Certain gamification elements such as meditation literature are also displayed due to recommendations in a special section of the application. For example, we assume that the user meditates in similar way as shown in Figures 5 and 6; such meditations are characterized by a stable heartbeat and a steady state during meditation. The application will count this as calm meditation and add the appropriate entry to the database (see Figures 25 and 26).

| | session_id [PK] integer | user_id [PK] integer | description character (255) | user_model character (255) |
|---|---|---|---|---|
| 1 | 101 | 1 | pulse_low/movement_low | Basic |
| 2 | 102 | 1 | pulse_low/movement_low | Basic |
| 3 | 201 | 2 | pulse_high/movement_low | Learn to meditate |

**Figure 25.** Example of data in the "Session" table.

| | x_axys<br>character varying (10) | y_axys<br>character varying (10) | z_axys<br>character varying (10) | heart_rate<br>character varying (4) | date_time<br>character varying (15) | session_id<br>[PK] character (1) |
|---|---|---|---|---|---|---|
| 1 | -2,129 | -0,226 | 1,315 | 79 | 13:51:34 | 101 |
| 2 | -2,129 | -0,186 | 1,275 | 72 | 13:52:38 | 101 |
| 3 | -2,109 | -0,186 | 1,295 | 72 | 13:53:35 | 101 |
| 4 | -2,09 | -0,186 | 1,315 | 73 | 13:54:45 | 101 |
| 5 | -2,109 | -0,206 | 1,295 | 73 | 13:55:37 | 101 |
| 6 | -2,148 | -0,185 | 1,315 | 72 | 13:56:40 | 101 |
| 7 | -2,129 | -0,186 | 1,315 | 71 | 13:57:41 | 101 |

**Figure 26.** Example of data in the "Sensor_Data" table.

Data on such a meditation will look similar to that shown in Figure 27. We observe a pulse that corresponds to the pattern of calm meditation; the data from the "Sensor_Data" table on the user's movement also correspond to calm meditation. In other words, meditation is characterized by pulse low/movement low. As a result of these data, the gamification element of the motivational model is activated, in our case "Badge with encouragement".

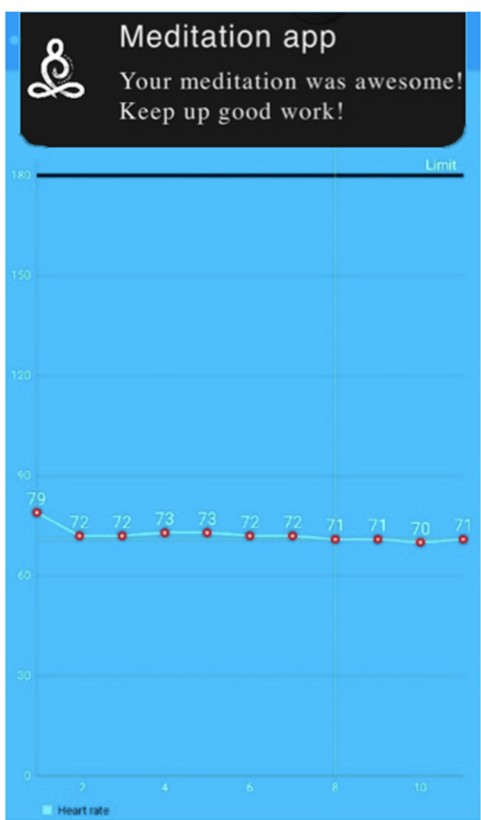

**Figure 27.** Example of the gamification element in application.

## 7. Conclusions

In the scope of the conducted research, we propose the meditation estimation model based on an analysis of literature, meditation videos from the constructed dataset, as well as consultations with meditation experts. We confirm the hypothesis that the external behavior of a person depends on his/her internal (mental) state, and state-of-the-art methods of artificial intelligence allow us to solve the problem of classifying human behavior based on the analysis of external signs obtained as a result of image analysis from the smartphone's camera and make conclusions about his/her internal state. We apply the modern methods of computer vision and pattern recognition to estimate the quality of the meditation process and provide the users with objective information about their practice. We constructed the meditation video dataset of meditating people that have been tracked by the

smartphone camera. The dataset contains more than 70 videos—each one is around 20 min—of 17 people including men and woman, experts and beginners. We collected meditation feedback by users for some videos and estimate all videos based on the developed parameters and estimation models. We built the neural network model that provides the possibility of detecting whether the human sits in the right meditation pose or not. This neural network model allows analyzing human movements and making correct recommendations to help people meditate better. We determine the main patterns of human behavior that are related to the meditation process, based on continuous measurement of the heart rate and acceleration parameters of the human body. We used popular wearable electronics devices to measure these parameters. We present the reference model for competence management of conscious exercise coaches for selecting the right coach for the right human based on his or her requirements and coach expertise. We built an audio guide recommendation model to recommend the most suitable actual audio guide. Using these models and the objective estimation model, we develop an approach to the system that will analyze human behavior, learn, and motivate them based on this analysis.

We propose using a smartphone with a camera to monitor the human meditation practice and estimate it. Furthermore, the human can use a wearable device for heart rate and movements estimation. In our experiments, we used a Xiaomi Mi band 3. At the moment, we are developing a prototype software that we are going to integrate into the mobile application that will be used for people during the meditation process. Distribution of the software will be implemented after the mobile application development.

**Author Contributions:** A.K. developed the reference model of the presented approach related to meditation practice estimation as well organized and annotated the accumulated Meditation Video Dataset. M.K. developed the competence management model and approach to the expert network platform. I.L. developed the proposed reference model together with Alexey Kashevnik. N.T. developed an approach related to meditation practice estimation based on human pose recognition. P.M. analyzed the meditation videos and developed the meditation estimation. E.R. developed the neural network model for meditation pose estimation. V.M. identified the main human behavior pattern and developed a mobile application for their recognition. N.S. developed a dynamic motivational model to stimulate the user of the meditation application to everyday proactive. I.R. conducted experiments related to human pose estimation for meditation scoring. All authors have read and agreed to the published version of the manuscript.

**Funding:** The competence-based model for a meditation coach has been developed in the scope of the Russian Foundation for Basic Research (project # 19-07-00670). The presented reference model has been developed due to the Russian State Research #0073-2019-0005. The neural network model for meditator pose estimation has been supported by the Government of the Russian Federation (Grant 08-08). Human physiobiological activity estimation based on wearable electronics was supported by FASIE.

**Acknowledgments:** We acknowledge all participants that have taken part in the Meditation Dataset Collection and especially Ella Zolotenkova for participating in the seminar and discussion of the proposed meditation estimation model.

**Conflicts of Interest:** The authors declare no conflict of interest.

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
