# Peer review of "Human Psychophysiological Activity Estimation Based on Smartphone Camera and Wearable Electronics"

_futureinternet, doi:10.3390/fi12070111_

Round 1
Reviewer 1 Report
This study is related to human psycho physiological activity estimation based on smartphone camera and sensors. The paper is interesting, but the structure is difficult for reading. I suggest restructuring this research with Introduction; Methods; Results; Discussion; Conclusions. The results must be compared with previous studies. The language should be revised by a native speaker.
Author Response
Thank you for the comments related to paper structure enhancement. We mentioned in introduction the following new structure.
Related work in the topic of heart and breathing rate correlation with psychophysiological activity as well as image recognition techniques for human activity detection is presented in Section 2. Also, we consider research work related to dynamic motivation strategies that can be applied to the task related to motivation of the human to everyday practice. Methods are presented in in Section 3, 4, and 5. We propose the reference model of the psychophysiological activity detection system in Section 3. Section 4 considers the meditation estimation approach and acquired dataset description. We propose user motivation model for psychophysiological activity in Section 5. Main results are presented in Section 6. Conclusion summarize the paper and contains main discussion the results got.

Reviewer 2 Report
The authors of this paper present a neural network model for classification of video frames in order to recognize the taken pose (human psychophysiological activity) and evaluate it. They are using smartphone and wearable devices to evaluate patterns of human behavior during meditation. The article is based on authors´ previous work (DOI: 10.23919/FRUCT48808.2020.9087356) as it is mentioned on the first page. However, link on this work in the references is missing.
The content of the article meets with the wider topics of the Future Internet journal.
Overall, the concept of the article is united. The abstract is adequate. The formal and scientific parts of the article meet with the basic requirements on a scientific journal paper. On the other hand, there is still open space to improve the overall quality of the article.
Consequently, the article needs major revision. Please, see my comments!
Comments:
- Article – the links on the references are not presented in order (see [8,10] is followed by [13])
- Article – the state-of-the-art is elaborated on good level. However, link on authors ‘previous work (DOI: 10.23919/FRUCT48808.2020.9087356) is missing. Why? The main differences between these works should be clearly presented as well as the main contribution of this work should be more highlighted.
- Article – the English grammar contains several minor typos (e.g. “we consider relationships beоtween breath rate and psychophysiological”; “Classifying Yoga Poses [16] It determines”; etc.). A careful back-check of the article is mandatory.
- Figure 4 – on this picture there is a camera and not a smartphone! Check it!
- Section 4 – (Figures 5/6) – it is not exactly clear that how many subjects were participated in the tests. Information about the subjects (on general level) should be presented.
- Section 3.2 – it is written: “We have used the meditation dataset described in the Section 6…”. Now we are in Section 3.2 – can you explain it?
- Figure 17 /18 – al subfigures are described in the text. Nevertheless, I recommend for the authors to give a name for all subfigures (recommended in all cases).
- Figure 21 – please, improve the visibility of this graph!
- Figure 22 – this figure and Fig. 7 in authors ‘previous work (DOI: 10.23919/FRUCT48808.2020.9087356). I recommend for the authors to redrawn this figure. Otherwise, problems with IEEE copyright can occur. The same problem occurs at Fig. 23. I also suggest slightly rewrite Sections 3.4 and 5 to reduce similarity with authors ‘previous work.
- Article – Section 3.4 is followed by Section 5. Where is Section 4?
- Section 7.1 – what is the meaning of “Oppositely, meditations done by beginners looks like completely different. The”?
- Article – sometimes, information about the used HW/SW components is missing. Next, it should be nice to put a link on the proposed models and codes – their share with scientific community can help the reproducibility of the tests and speed up future research works.
- Sections 7 / 8 – if it is possible, then the features of the proposed approach with the SOTA solutions should be better compared.
Author Response
Dear reviewer, thak you so much for the interesting comments. We summarise them in the table and provide detailed answers.

Round 2
Reviewer 1 Report
There are a high number of images, but it was improved as requested.
Author Response
Dear reviewer,
Thank you for you decision. Attached you can find answers to another reviewer.
Alexey Kashevnik

Reviewer 2 Report
Many thanks for the explanation letter!
In general, authors´ respond on my comments is adequate. The article has been improved. Just one thing. Next time, please, upload a revised manuscript, where the changes in the text are highlighted in, for instance, yellow color. The uploaded manuscript in the option “tracking of changes” has very low readability. Thank you for your understand!
After its check, I have the following comments/questions.
Comments:
- Article – the are still some minor typos in the English grammar (e.g. “… in in Section 3,4 and 5.”).
- #6 – only one person has been participated in the tests! In my opinion it is not enough. The outputs of the study cannot be relevant. Please, extend the outputs of your experiment with a conclusion, based on the “activities” of (minimally) 5 subjects.
- #9 – some figures still have bad visibility (see figure on pp.23)
- #13 – here I consider the method – to monitor and evaluate the behavior of a subject during his activities…
Author Response
Dear reviewer,
Thank you so much for you comments. We have fixed the paper and highlighted changes by yellow.
Alexey Kashevnik

Round 3
Reviewer 2 Report
The article has been improved.